



# The effect of anthropogenic heat emissions on global warming

Dimitre Karamanev

Department of Chemical and Biochemical Engineering, University of Western Ontario, Ontario, Canada, N6A 5B9
*Correspondence to:* D. Karamanev (dkaraman@uwo.ca)

**Abstract.** The use of different primary energy sources in human society has led to two major polluting emissions in the environment: energy (mostly heat), and chemical substances (mostly carbon dioxide). In this paper, a new approach, based on the similarity between sensible heat and $CO_2$ transfer properties, was used to determine the effect of anthropogenic heat release on the global air temperature. The total global anthropogenic emissions of sensible heat were divided into two separate streams:

directly transferred to: (1) water and land, and (2) to the atmosphere. The direct emissions of heat to the atmosphere during the industrial era (years 1850-2018) were determined and their effect on the change of global atmospheric temperature was calculated. The global atmospheric temperature increase caused by anthropogenic heat emissions was estimated. The resulting calculations showed that at least half of the actual atmospheric temperature rise recorded during the last 170-year period, was due to the anthropogenic heat release. These results suggest that the temperature change of the atmosphere (global warming)

is strongly affected by anthropogenic heat emissions.

## 1. Introduction

Traditionally, there have been two known types of living organisms defined by the primary form of life energy they consume – phototrophs and chemotrophs, which use the energy of light and chemical energy, respectively (Starr, 2001). In addition, a third type of organism was recently discovered. It was named an electrotroph due to its ability to consume directly electrical

energy (D, 2003). In all three groups of organisms, the various types of energy needed to sustain life (i.e. mechanical, thermal, electrical, chemical) are produced via conversion of the three primary forms of life energy, which takes place inside the living organisms (*in vivo*). The main exception are humans, who in addition to the in vivo energy conversions, make use of energy conversions outside of their bodies (*ex vivo*) by producing forms of energy not directly needed for life. Historically, the first and still the most important ex vivo energy conversions practiced by humans is the conversion of chemical to heat energy (fuel

oxidation), or the command of fire. Millions of years later, people began exploring the oxidation of alternative fossil-based fuels.

The process of chemical fuel oxidation involves two significant types of emissions to the environment: *mass* (primarily carbon dioxide and water vapor) and *energy* (mostly as sensible heat, plus some electromagnetic energy in the visible and infrared spectrum). With the continuing industrialization of society dramatic increases in both carbon dioxide and heat emissions have

been taking place on a global scale. Since the beginning of the $20^{th}$ century, the $CO_2$ concentration in the atmosphere has



reached such high levels that it has been purported to be the primary cause of the dramatic increase of atmospheric temperature observed, commonly referred to as "global warming". To prove experimentally the effect of $CO_2$ as a greenhouse gas on atmospheric temperature, one should be able to determine the difference between the incoming and outgoing radiation energy (radiative flux imbalance, RFI) at the top of the atmosphere (von Schuckmann et al., 2016). The energy balance on Earth shows

that when RFI=0, the long-term average global surface temperature does not change. However, when RFI is non-zero, the Earth temperature changes. For example, if the incoming radiative flux is higher than the outgoing one, the temperature on Earth will increase due to the resulting energy accumulation. It was calculated that in order for the RFI to produce the currently observed global warming, the energy imbalance at the top of the atmosphere would need to be so small, that it would fall much below the margin of error of individual flux measurements (Stephens et al., 2012). Therefore, there is still no direct proof that

carbon dioxide emissions cause global warming. In order to assess the impact of greenhouse gases (such as $CO_2$) emissions on atmospheric temperature increase, mathematical models have been widely used (Stocker et al., 2013). Unfortunately, the processes in the atmosphere are extremely complex and their modelling requires the use of many variables and constants that are either assumed or fitted. As a result, the Intergovernmental Panel for Climate Change concluded in its latest report (Stocker et al., 2013) that the probability of global warming being caused by increasing atmospheric $CO_2$ concentration is 95%. While

this number seems high, it is still far from 100%. There have been many projected events of different nature with a higher expected probability that never actually occurred.

As mentioned above, there are two main anthropogenic emissions to the environment: mass and energy, which are primarily comprised of carbon dioxide and heat, respectively. While the global effect of $CO_2$ on the environment has been studied extensively, anthropogenic heat accumulation has been studied mostly on a local scale in relation to "urban heat islands" (Chen

et al., 2016; Dong et al., 2017). The global effect of direct anthropogenic heat release into the atmosphere has been studied in several papers, and it has been found that the amount of anthropogenic heat flux comprises only approximately 1% of greenhouse gas forcing (Chen et al., 2016; Crutzen, 2004; Flanner, 2009). However, in my view, there are some problems in these models, which are outlined below.

This work analyzes the *direct heat release into the atmosphere* by human society during the industrial era (starting at year

1850). An innovative approach based on the similarity of sensible heat and $CO_2$ properties in the atmosphere, has been used. The main assumptions in the simple model proposed here, are different from these used in the previously published climate models. The calculated atmospheric temperature increase brought on by the anthropogenic heat emissions is compared to the recorded global atmospheric temperature increase.

## 2. Main assumptions of the model

One of the major assumptions made in this paper is the similarity between properties of sensible heat energy and $CO_2$, both emitted by human society. These similarities are listed below:



- The releases of heat and $CO_2$ by fuel oxidation are, as an average, stoichiometric;
- Their spatial and temporal distributions due to convection in the atmosphere are similar;
- Their transfer in boundary layers is similar;
- They both can be accumulated in atmosphere;
- The top of the atmosphere is impermeable to both $CO_2$ and sensible heat.

The main difference between sensible heat and $CO_2$ properties is in their interaction with land mass and sea surface, and also the radiation processes in the case of heat transfer.

These similarities are used below for the analysis of current anthropogenic heat flux (AHF) climate models.

The current climate models involving AHF show that the global average AHF represents approximately only 1% of greenhouse gas forcing. However, in my view, there are several significant issues with these models:

- In the AHF climate models it was assumed that anthropogenic heat release is a "heat flux" (AHF). By definition, heat flux is the rate of energy flow per unit of Earth surface area in terms of $W/m^2$, i.e. it is a two-dimensional process. While radiation processes and their derivatives (natural sensible and latent heats) are best described as
energy fluxes in two dimensions (in $W/m^2$), in my view the anthropogenic heat release has a three-dimensional nature expressed as $J/m^3_{atmosphere}$ or alternatively $J/kg_{atmosphere}$, similar to anthropogenic release of carbon dioxide ($kg_{CO2}/m^3_{atmosphere}$ or $kg_{CO2}/kg_{atmosphere}$). This is due to the similarity between the properties of $CO_2$ and sensible heat energy emissions.
- The AHF does not take into account the cumulative effect of anthropogenic heat release. However, part of the
anthropogenic heat is obviously accumulated over time in the atmosphere. In this paper the cumulative effect of heat release is taken into account by Eq. 10.
- It was assumed that "all non-renewable primary energy consumption is dissipated thermally in Earth's atmosphere" (Flanner, 2009). However, this paper (Table 1) shows that only 40% of the primary energy used by humans is transferred directly to the atmosphere. The rest is transferred directly to bodies of water and in some cases to land,
and is assumed to have an insignificant effect on atmospheric temperature.
- In AHF models, anthropogenic heat was considered to have been added as "dry static energy to the lowest layer of the atmosphere" (Flanner, 2009). The data in Table 1 of this paper shows that part of the anthropogenic heat is released to the atmosphere high above Earth's surface, for example by airplane jet engines and as latent indirect heat release by electrical power plants.



The four problems addressed above were solved in this paper. As a result, it is shown here that anthropogenic heat release is responsible for nearly 50% of the atmospheric temperature rise, which contrasts the previous climate models that estimated the contribution at merely 1%.

### 3. Analysis of anthropogenic heat emissions

It seems that there is no data in the available literature on the amount of anthropogenic heat transferred *directly to the*
*atmosphere.* Usually it is assumed that *all* the primary non-renewable energy used by human society is converted to heat, transferred directly to the atmosphere (Dong et al., 2017; Flanner, 2009; Varquez et al., 2021), which as shown above, may not be correct. In order to estimate the amount of anthropogenic heat energy introduced directly into the atmosphere, the following analysis and calculations were performed.

Anthropogenic sensible heat is transferred primarily to water, land mass and atmosphere. Heat transfer to ice and vegetation
is considered insignificant for the purposes of this analysis. In this work only anthropogenic heat, transferred directly to the atmosphere will be dealt with. The main reason for that is the substantial difference between the volumetric heat capacity of air, as opposed to these of land mass and water. The volumetric heat capacity of air at sea level ($0.0012$ J/m$^3$K) is three orders of magnitude lower than those of fresh and ocean water ($3.0 - 4.18$ J/m$^3$K) or soil ($0.3$-$3.1$ J/m$^3$K) (Oke, 2002). Therefore, when a certain amount of heat is transferred to either water or soil, it will result in a temperature increase that is thousands of
times lower than in the case where the same amount of heat is transferred to air having the same volume (please see Eq. 1 below). Even taking into account that the atmospheric heat is assumed to be much better distributed in the entire atmosphere due to various convection processes, there is still a significant temperature difference between the atmosphere from one side and the land and water from the other. Furthermore, the small increase in water temperature would not significantly impact evaporation. Hence, only anthropogenic heat introduced directly into the atmosphere will be considered here.

**Precision of the measurements.** There is no information in the available literature on the anthropogenic heat emission to the atmosphere. The available data on the overall heat emissions from different energy conversion technologies is not related *directly* to the heat transfer to the atmosphere. For example, as shown below, the direct heat losses of internal combustion engine automobiles are approximately 71%, while the transfer of heat to the atmosphere, when including indirect heat emissions, are 93% of the fuel energy. That is because of the transformation of significant amount of the "useful" mechanical
energy (22%) to atmospheric heat as a result of different friction processes. While it was possible to estimate quite precisely the heat emissions to the atmosphere for some types of energy conversions, in other cases they were wild guesses. The precisions of the individual heat emission estimates to the atmosphere are shown in Table 1. The total errors are shown as error bars in Fig. 2. It should be noted that the atmospheric heat releases by emitters, determined with lower error (below 10%, such as the automobiles, jet engines and heat engines for electrical power generation), account for more than half of the total
emissions. However, the fraction of the high-error emitters (above 30% error, such as the industrial use of oil and in the railway



transport), account for only 1/5 of the total. Therefore, the overall determination of the atmospheric heat releases is relatively precise.

The amounts of anthropogenic sensible heat energy introduced into the environment, and specifically into the atmosphere, as a result of conversion of primary to final forms of energy are shown in Table 1.

**Electric power plants.** Thermoelectric power plants, particularly nuclear, coal and natural gas ones, are among the world greatest emitters of heat to the environment. All of these power plants use heat engines (turbines) to convert thermal energy into mechanical energy. Since the efficiency of conversion of heat to mechanical energy in turbines is between 30 and 50%, at an average of 33% (Smil, 2019), an enormous amount of waste heat is released to the environment, equal to 67% of the entire energy content of fuel. Most of this heat is released during the process of condensation of steam leaving the turbine.
Three main technologies are used to transfer waste heat from thermal power plants to the environment: indirect (recirculating), dry cooling, and direct (once through) (Meldrum et al., 2013).

Indirect heat release is a result of evaporative water cooling in cooling towers. The heat released during this process is converted to latent heat of water vaporization. The vapor produced, after rising upwards in the atmosphere and subsequently precipitating, converts latent heat to the equivalent amount of sensible heat, which is transferred to the air. Therefore, the
amount of heat emitted by thermal power plants' indirect cooling process will be considered here as part of the overall atmospheric heat input.

Dry cooling technology is based on the process of heat transfer from a power plant to the atmosphere through the use of heat exchangers similar to radiators in automobiles. Therefore, it will also be considered here as an atmospheric heat input.

In the case of direct heat removal, waste heat from the power plant is transferred to fresh or ocean water. Since this heat is not
transferred to the atmosphere, it will not be considered here.

It has been estimated that direct heat removal is used in the generation of 89% of electricity by thermoelectric power plants worldwide (Lohrmann et al., 2019). Therefore, 11% of the total heat loss of coal, natural gas and biofuel-based thermoelectric power plants worldwide is released into the atmosphere by indirect and dry cooling, and the rest (89%) is transferred to water bodies. In the case of fossil fuel power plants, global energy statistics designate the chemical energy provided by these fuels
as "fuel energy supply" (Fig. 1). At an average efficiency of 33%, the heat energy transferred to the environment is equal to 67% of the total fuel energy supply. Therefore, the emissions of thermal energy to the atmosphere by fossil fuel and biofuel power plants is equal to 0.67*0.11= 0.074, or 7.4% of the energy content of fuels used for electric power generation.





However, in the case of nuclear power plants, the amount of "global nuclear energy supply" is usually reported as the amount of electrical energy generated by the plants, and does not include any waste heat (IEA, 2020). Since nuclear power plants have

approximately 33% efficiency, the amount of waste heat is double the amount of electrical energy generated. Eleven percent of the waste heat is released into the atmosphere globally. Therefore, the total amount of heat released into the atmosphere by nuclear power plants is equal to 2*0.11=0.22, or 22% of the "world nuclear energy supply" (IEA, 2020) (Fig. 1).

Atmospheric heat release by hydroelectric power plants is negligible, and is assumed to be zero.

**3.1. Oil**

**Transport.** Another major source of anthropogenic heat input to the atmosphere is the transportation sector. There are two main types of engines used: internal combustion and airplane jets.

*Road and rail transportation.* The average energy efficiency of internal combustion engines is estimated to be 28% (National Network for Energy Efficiency, 2010). The remaining chemical energy of the fuel is transferred directly to the atmosphere by means of exhaust gases and heat exchange between the metal parts of the engine (such as the radiator) and the surrounding air.

Part of the mechanical energy generated by the engine (28% of the chemical energy of fuel) is used to overcome air friction, as well as mechanical drag forces in the drive train and in the brakes. The entirety of this energy is further transferred as sensible heat to the atmosphere. The only types of energy that are not transferred to the atmosphere as sensible heat are radiation heat (considered insignificant here) and the rolling resistance of the wheels. There are some notable differences between automobiles (including light trucks and vans) and large transport trucks:

In automobiles, rolling resistance is around 7% (National Network for Energy Efficiency, 2010). Therefore, automobiles transfer 93% of the chemical energy of their fuel to the atmosphere as sensible heat. Automobiles account for 62 percent of global surface transportation energy consumption (Anon, 2016).

In large transport trucks, the rolling resistance is 17% (National Network for Energy Efficiency, 2010). Therefore, transport trucks transfer 83% of the chemical energy of their fuel to the atmosphere. Trucks account for 36 percent of global

transportation energy consumption (Anon, 2016).

*Rail transportation.* Approximately 2% of global surface transport is comprised of railways (IEA, 2020). It is assumed that rail transport converts 97% of the chemical energy of its fuel to the atmosphere as sensible heat.

*Sea navigation.* There are two major types of sea vessels: transport and cruise ships. Both of them primarily use internal combustion engines. Since cruise ships account for only a small amount of the entire fleet, we will bundle them together with

transportation ships. Ship engine cooling is achieved by circulating ambient water (Prasad Sinha, 2015), and therefore is not transferred to the atmosphere. In addition, practically all the mechanical power of the propeller is also transferred to water

(Landeka and Radica, 2016). Therefore, the energy transferred to the atmosphere is released mainly by the stack, which accounts for 15% of the fuel energy (Prasad Sinha, 2015).

*Turbine jet engines.* The energy of fuel (kerosene) used in aircraft jet engines is converted directly to heat (heat loss) at a rate of approx. 75% (NAS, 2016). The rest of the fuel energy is converted to mechanical energy, which is ultimately also converted to heat mostly due to friction processes and ends up in the atmosphere. Therefore, the heat energy released into the atmosphere by a turbojet engine is approximately 100% of the chemical energy of fuel.

**Non-energy industrial use of oil.** In this application petroleum is used primarily as feedstock in chemical and other industries, as well as for road paving. It is assumed that there are no significant heat emissions due to the associated chemical transformations.

**Industrial use.** Heat emissions to the atmosphere resulting from industrial oil use are assumed to be similar to those from power plants, i.e. 7.4%.

**Commercial and residential use.** Use of oil products in this context is mostly for space heating. It has been shown that the average heat release from buildings into the atmosphere is 50% (Simon, 2017).

### 3.2. Coal

**Electricity generation.** Globally, most of the coal is used in thermoelectric power plants. As mentioned above, the heat release into the atmosphere is 7.4% of the chemical energy of coal.

**Industrial use**. The second most significant use of coal is in industry, where it is chiefly employed for the production of iron and cement. The reported thermal balance of coal-based iron production facilities indicates that the heat emission to the atmosphere is approximately 20% (Lu et al., 2019). There are different technologies for cement production, and it is estimated that the heat release to the atmosphere is 30% (Khurana et al., 2002; Kumar Verma et al., 2020). The overall heat emissions of industrial processes to the atmosphere was taken as 25%.

### 3.3. Natural gas

**Electricity generation.** As mentioned above, natural gas power plants emit 7.4% of the chemical energy of their fuel to the atmosphere as a heat loss.

**Industrial use.** It is assumed that the most ubiquitous use of natural gas in industry is for the electricity generation, where the heat release to the atmosphere is 7.4%.



**Commercial and residential use.** In this application natural gas is used primarily for space heating. As in the case of space heating by oil, the heat release to the atmosphere is approximately 50%.

Most of the natural gas used in industry for **non-energy** purposes functions as chemical feedstock. Therefore, it is assumed that the resulting atmospheric heat emissions are negligible at 0%.

### 3.4. Biofuels and waste

**Commercial and domestic** use of biofuels is mostly for space heating, at 50% atmospheric heat emissions.

**Electricity generation** involving biofuels and organic waste is performed using thermoelectric power plants, where heat 210 emissions to the atmosphere are 7.4%, as mentioned above.

The use of biofuels and organic waste in **industry** is assumed to release 20% of heat to the atmosphere.

### 3.5. Nuclear energy

Nuclear energy is used exclusively in thermoelectric power plants. As mentioned above, in this case the heat release represents 22% of the world's nuclear energy supply.

### 3.6. Hydro power

Electricity generation by hydroelectric power plants is not associated with any significant heat release to the atmosphere, and is assumed here to be zero.

### 3.7. Electricity

Among all the types of energy shown in Table 1, electricity is the only form of energy that is not a primary energy source. 220 However, after being produced, there are two main types of thermal losses associated with electricity: first as a result of ohmic resistance during electricity transmission and second, due to the transformation of electricity to the final forms of energy on the consumer end.

The heat released during the conversion of electricity to the **final useful type of energy** such as mechanical (in motors), thermal (space heating), electromagnetic (lighting) and so on is estimated next. 48.4% of electricity generated worldwide is 225 used for commercial and residential purposes, where it was assumed that 50% of it is released to the atmosphere (Simon,





2017). The rest (51.6%) is used in industries and other applications. It is roughly estimated that 30% of that energy is released to the atmosphere. Therefore, the total atmospheric heat input from the final use of electricity is estimated at 40%.

Thermal losses associated with electrical power **transmission** comprise 8% of global electrical energy generation (Jackson, 2015). These losses are sustained in the form of sensible heat transferred directly to the atmosphere.

**3.8. Heat generated by living organisms**

The total amount of life forms on Earth is estimated to be approximately 550 gigatons in terms of carbon (Barnosky, 2008). Of these, only chemotrophs living on the surface of Earth (such as humans) or in the air (Barnosky, 2008) are considered here as significant sources of heat emission to the atmosphere. It has been estimated that the global amount of chemotrophs has increased by less than 1% in the last 120 years (Bar-On et al., 2018), and therefore, the change in their heat release to the

atmosphere during that period is considered negligible.

**4. Total anthropogenic heat release into the atmosphere**
The anthropogenic heat release into the atmosphere was determined by multiplying each specific energy input by the fraction of that energy going to the atmosphere as heat (Table 1). In summary, the total amount of energy used globally in 2018 was 13.55 gigatons of oil equivalent (Gtoe), and out of that 5.39 Gtoe (40%) was released to the atmosphere as sensible heat.

Table 1 also shows each primary energy source's contribution to total atmospheric heat release during year 2018. It is assumed that this contribution did not change significantly over time since 1920s. It was used to calculate atmospheric heat emissions by each type of primary energy source during the period 1920-2018. Before 1920, the main primary energy sources were biofuels and coal, the latter mainly used in steam engines (stationary and railways), steel production and space heating. Because of the absence of statistical data, it was assumed that a total of 80% of the primary energy used at that time period was

transferred to the atmosphere as heat.

The amounts of different types of primary energy used globally by humans between the years 1800 and 2018 are shown in Fig. 1. The underlying data was obtained from Smil (Smil, 2019) for primary energy sources prior to 1960; Smil (Smil, 2017) for electricity generation prior to 1960 and IEA (IEA, 2020) for both primary energy sources and electricity for the period 1970-2018. Since the energy use from traditional biofuels globally was fairly constant between the years 1800 and 1970 (Fig.

1), that amount (an average of 0.573 Gtoe per year) was subtracted from the total for all the years between 1850 and 2018.



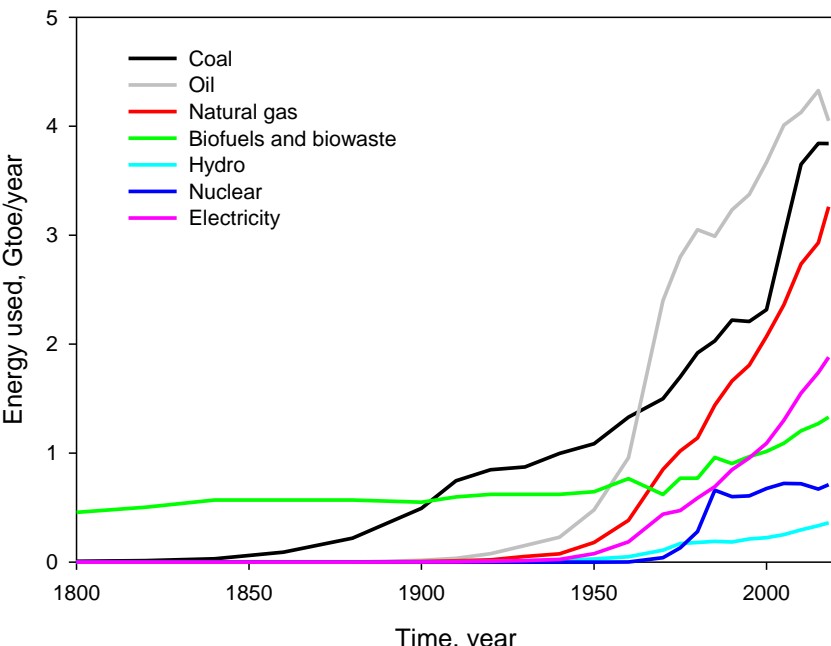

**Figure 1:** Annual global use of different primary energy sources.

## 5. Calculation of temperature rise of the atmosphere

Analysis of heat emissions in the period 1850-2018 (Fig. 1 and Table 1) has revealed that a total of $1.17*10^{22}$ J was released into the atmosphere as a result of human activity. Let's assume first that there were no losses of heat from atmosphere during that period. While that is impossible, the calculation will show the total potential thermal effect of the anthropogenic heat input. It is well known that the relationship between the changes in sensible heat and temperature is:

$$Q_{in} = mC\Delta T \tag{1}$$

Where $Q_{in}$ is the heat input to the mass m, and $\Delta T$ is the change of temperature due to the heat input. The total mass of the atmosphere is $m=5.1\cdot10^{18}$ kg (Budyko, 1982). The heat capacity of air at atmospheric pressure is $C=1000$ J/kg°K and does not change significantly with decreases in pressure (i.e. with the altitude) (Green, 2008). Using these numbers, the atmospheric temperature increase since beginning of the industrial revolution, assuming no heat loss, would be 2.3°C! Therefore, the anthropogenic heat release has a significant theoretical potential of changing the atmospheric temperature.





As mentioned above, there is always loss of some of the heat added to the atmosphere. There are two main possible losses: (1) downwards to the land and ocean, and (2) upwards to space as long wave radiation. Since the average Earth temperature is higher than the average atmospheric temperature just above the Earth, the increase of the air temperature would decrease the temperature difference between the land and atmosphere, and therefore, the heat transfer would slightly decrease. Here that change in heat transfer rate will be assumed to be insignificant. Therefore, the main heat loss of the anthropogenic heat input

to the atmosphere would be due to the long wave radiation to space.

The energy balance of sensible heat introduced to the atmosphere by society ($Q_{in}$) can be written as:

$$Q_{in} = R + Q_{atm} \tag{2}$$

and therefore:

$$dQ_{in} = dR + dQ_{atm} \tag{3}$$

where R is the outgoing radiative emission leaving the atmosphere and $Q_{atm}$ is the amount of sensible heat remaining in the atmosphere. There are several analytic, empirical and semiempirical equations, proposed to calculate R as a function of the atmospheric temperature. The most procise of these models seems to be the Budyko (Budyko, 1982) equation:

$$R = a + bT - (a_1 + b_1T)n \tag{4}$$

where $a$, $a_1$, $b$ and $b_1$ are constants, T is the air temperature near the Earth surface in degrees Celsius and n is the atmospheric

cloudiness, measured in the fraction of a unit. Taking a temperature differential:

$$\frac{dR}{dT} = b - b_1 n \tag{5}$$

therefore:

$$dR = (b - b_1 n)dT \tag{6}$$

In general, the relationship between sensible heat and temperature is given by:

$$dQ = mCdT \tag{7}$$

Combining Eqs. 3, 6 and 7:



$$dQ_{atm} = mCdT = dQ_{in} - dT(b - b_1n) \tag{8}$$

Rearranging Eq. 8, it can be shown that:

$$dT = \frac{dQ_{in}}{mC + b - b_1n} \tag{9}$$

The values of m and C are shown above. The initial temperature $T$ around year 1850 was set to 0 (Fig. 2), which allowed for the calculation of the atmospheric temperature rise $\Delta T(t)$ between the years 1850 and 2018. The constants $(b-b_1n)=1.8 \cdot 10^{22}$ J/°K (Budyko, 1982), calculated for the surface of the entire planet. The average planetary cloudiness was reported to be between 0.49 and 0.67 (Mokhov and Schlesinger, 1994). The variation is mostly due to different methods of measurement, but for each method the time variation since 1970s was small, around 1-3% (Mokhov and Schlesinger, 1994; Roscow and Schiffer,

1999; Stubenrauch et al., 2013). Here the average cloudiness was chosen at n=0.50, as that value was used in the derivation of Eq. 4 (Budyko, 1982). Therefore, $1/(mC + b - b_1n)=0.438 \cdot 10^{-22}$ °K/J.

After integration of Eq. 8, the temperature rise as a result of the anthropogenic heat release to the atmosphere between years 1850 and $t$ can be calculated as:

$$\Delta T = T(t) - T(1850) = 0.425 \cdot 10^{-22} Q(t) \tag{10}$$

The temporal data for Q(t) was obtained from Fig. 1 and Table 1.

As a first approximation, it was assumed that anthropogenic heat input is equally distributed throughout the entire Earth's atmosphere (Jacob, 2000). Since $CO_2$ is well mixed within the atmosphere (Stocker et al., 2013), using the heat and mass transfer analogy, it can be assumed that the sensible heat released by human activity, is also well mixed. In addition, it has been shown that the local anthropogenic heat emissions are quite well distributed vertically in the atmosphere (Chen et al.,

305    2016).

The calculated atmospheric temperature increase since 1850 is shown in Fig. 2. The Figure also shows the observed annually averaged land-mass global temperature change since 1850 (Anon, 2022). A close relationship can be seen between the actual recorded temperatures and the ones calculated on the basis of anthropogenic heat emissions to the atmosphere (Eq. 9).

These results suggest that the heat introduced into the atmosphere by human activity is extremely important for the atmospheric

temperature increase during the industrial era, and accounts for approximately half of the temperature rise. The other half is due to the greenhouse gas effect.





**Table 1.** The global primary energy use (IEA, 2020) and anthropogenic heat emissions in 2018.

| Type of primary energy | Primary energy flow (Gtoe/yr) | Primary energy users | Energy used (Gtoe/yr) | Sensible heat release % | Sensible heat release (Gtoe/yr) | Average heat release % | Estimated error of % release |
|---|---|---|---|---|---|---|---|
| Oil | 4.05 | Automobiles | 1.280 | 93 | 1.20 | 67% | 10 |
| | | Trucks | 0.743 | 83 | 0.62 | | 10 |
| | | Rail | 0.041 | 97 | 0.06 | | 40 |
| | | Aviation | 0.336 | 100 | 0.336 | | 5 |
| | | Navigation | 0.275 | 25.5 | 0.070 | | 20 |
| | | Industry, non-energy | 0.648 | 7 | 0.045 | | 50 |
| | | Industry, energy | 0.292 | 7.4 | 0.022 | | 20 |
| | | Commercial, residential | 0.441 | 80 | 0.353 | | 20 |
| Coal | 3.84 | Electricity generation | 2.460 | 7.4 | 0.182 | 14% | 5 |
| | | Industry | 0.798 | 20 | 0.160 | | 30 |
| Natural gas | 3.26 | Electricity generation | 1.30 | 7.4 | 0.096 | 22% | 5 |
| | | Industry | 0.60 | 11 | 0.066 | | 40 |
| | | Commercial, residential | 0.70 | 80 | 0.56 | | 20 |
| | | Non-energy | 0.19 | 0 | 0 | | +10 |
| Biofuels and waste | 1.33 | Commercial, residential | 0.72 | 80 | 0.576 | 47% | 20 |
| | | Electricity generation | 0.20 | 7.4 | 0.015 | | 5 |
| | | Industry | 0.20 | 20 | 0.04 | | 50 |
| Nuclear | 0.71 | Electricity generation | 0.71 | 22 | 0.156 | 22% | 5 |
| Hydro | 0.36 | Electricity generation | 0.36 | 0 | 0 | 0 | 0 |
| Electricity* | 1.73 | Final types of energy | 1.73 | 40 | 0.69 | 40% | 50 |
| | 1.88 | Transmission losses | 1.88 | 8 | 0.15 | 8% | 5 |
| Total** | 13.55 | | 13.10 | | 5.39 | 40% | |

*Not a primary energy source.

**The total energy used is less than the total primary energy input because some minor inputs, emitting insignificant amount of heat, are not listed in the table. Examples: wind and solar electricity generation.





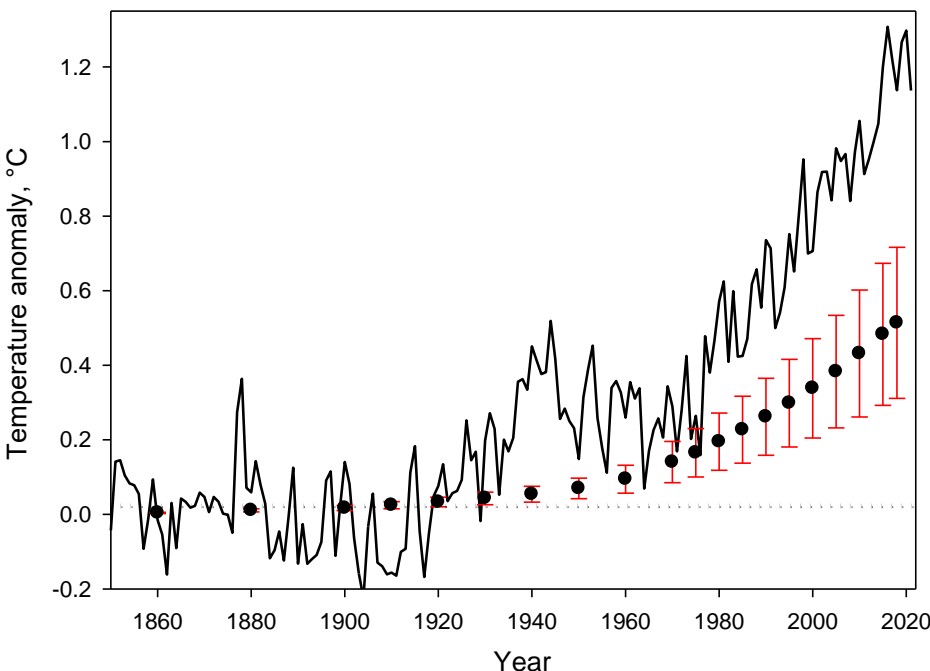

**Figure 2:** The atmospheric temperature change since 1850. Line: actual data; black points: data calculated using Table 1 and Eq. 10.

**6. Radiative flux imbalance at top of the atmosphere due to anthropogenic heat input**

As shown above (Eq. 2), part of the anthropogenic heat entering the atmosphere remains in it, while another part is radiated to space. For year 2018, the amount of long wave radiation leaving the planet due to anthropogenic heat release, calculated by Eq. 6, was 0.02 W/m². It should be noted that while the RFI is positive in the case of greenhouse gas effect, it is negative as a result of anthropogenic heat input to the atmosphere.

**7. Historical relationship between anthropogenic mass ($CO_2$) and energy (heat) emissions**

As mentioned above, there is expected to be a close relationship between anthropogenic carbon dioxide and heat emissions to the atmosphere. As the proportion of fossil energy in the entire global energy mix has been higher than 70% since the 1950s, it is expected that amount of anthropogenic carbon dioxide emissions would correlate with the amount of heat emitted to the atmosphere. Figure 3 shows this relationship. It can be seen that carbon dioxide emissions are almost proportional to atmospheric heat input. The difference in the slopes between and after 1970 is probably due to the change in the ratio between different primary energy sources. Therefore, the expectation is that any future reduction in atmospheric heat emissions will be





closely correlated with a reduction in the carbon dioxide emissions. For example, the transition from fossil fuel based to renewable power generation will have similar positive effect on both atmospheric heat emissions and the $CO_2$ emissions. However, there are some exceptions. In recent years, thermoelectric power generation authorities (which employ nuclear, fossil

and bio- fuels) are quite interested in moving from water cooling to air cooling, in order to reduce the load on water resources (Lohrmann et al., 2019). Unfortunately, this move would significantly increase heat emissions to the atmosphere and therefore would inevitably result in an atmospheric temperature increase. In addition, according to the above findings, since nuclear and biofuel power plants emit a similar amount of heat to the atmosphere as fossil fuel plants do, these three types of power plants present a similar global warming potential.


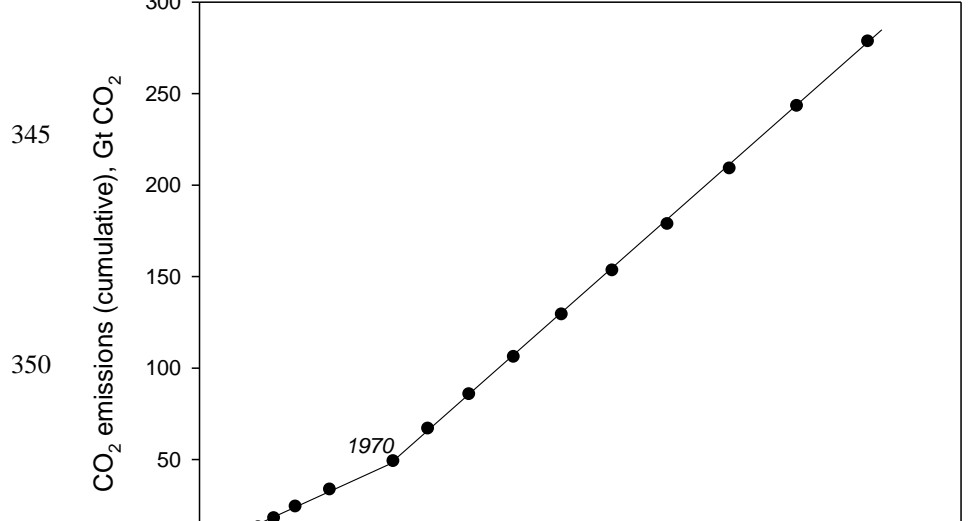

**Figure 3:** Relationship between the anthropogenic mass ($CO_2$) and energy (heat) emissions to the atmosphere.

There is another interesting effect of the relationship between the anthropogenic heat emission and atmospheric temperature. Let's assume that 100% decarbonization of the primary energy used by society is achieved by wind and solar electricity generation. In that case there will be still a significant heat emission to the atmosphere because the majority of the energy used will ultimately end up as heat, and much if it will be emitted to the atmosphere as sensible heat. It was estimated that the anthropogenic heat input to the atmosphere at 100% decarbonized economy will be 54% of the current atmospheric heat



release. The heat released by wind turbines and photovoltaic panels was not included in the calculations. Therefore, even at 100% decarbonization, the atmospheric temperature rise will be still approximately ¼ of the current one.

## 8. Conclusions

The atmospheric temperature rise due to anthropogenic *heat emissions directly to the atmosphere* was calculated for the period 1850-2018 on the basis of a three-dimensional heat distribution approach. The calculations were based on a heat and mass
transfer analogy. It was shown that the temperature increase of the atmosphere due to anthropogenic heat emissions is approximately half of the global land-sea temperature rise measured experimentally (global warming). Therefore, it can be concluded that anthropogenic heat emissions have a significant effect on global warming. On the basis of the above calculations, in order to manage global warming, a strong emphasis must be placed on reducing not only $CO_2$ emissions, but also heat emissions to the atmosphere. In most cases, but not always, a heat emission reduction is closely associated with a
reduction in $CO_2$ emissions.

It was shown that even at 100% decarbonized economy, there will be still ¼ of the current atmospheric temperature rise due to the anthropogenic hat release.

From energy point of view, the main difference between humans and the rest on the living world is the fact that humans use
both in-vivo and ex-vivo energy conversions, while the other forms of life use exclusively in-vivo energy conversions.

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

Author contribution: D. Karamanev contributed to 100% of the paper

Competing interests: No competing interests are claimed