# Peer review of "The effect of anthropogenic heat emissions on global warming"

_EGUsphere, 2022_

## Referee Comment (RC1)

**Review report of the manuscript "The effect of anthropogenic heat emissions on global warming," manuscript no. egusphere-2022-5**

This study calculates atmospheric temperature rise from 1850 to 2018 due to anthropogenic heat emissions to the atmosphere, based on a three-dimensional heat distribution approach. This study concludes that the atmosphere's temperature is half of the observed global land-sea temperature rise, and anthropogenic heat emissions significantly affect global warming.

This study raises very crucial issues but with a weak approach. The introduction section lacks a clear explanation about why this study is necessary and what are gap areas this study attempts to fill. Line 53 depicts that "there are problems in models used in the previous studies," but it does not explain the details. Line 39 says, "there is no direct proof that CO2 causes global warming" however, it does not explain this statement. Line 43 mentions the latest IPCC report but cites a 2013 paper. The introduction section should be constructively revised and clearly explain the gap areas and the detailed explanation of the need for this study.

The main assumptions of the model described in lines 62-66 are very theoretical. A details explanation is required for each assumption. Line 63 "the spatial and temporal distribution of heat and CO2 due to convection is similar in the atmosphere". Here do you mean latent heat or sensible heat? Sensible heat is energy that moves from one system to another, changing its temperature. The transport and distribution of CO2 throughout the atmosphere are controlled by the jet stream, large weather systems, and other large-scale atmospheric circulations. A detailed explanation is required for this assumption. Also, the authors consider only CO2 in their assumptions; however, black carbon is found to influence the sensible heat very differently compared to other aerosols and greenhouse gases due to its strong atmospheric absorption

There are confusing statements in lines 67-69, e.g., the "main difference"…." The similarities"……. Please re-write with a clear explanation.

Re-arrange section 3; there are un-numbered sub-sections, e.g., precision measurements and electric power plants. It is better to merge all sub-section in section 3. The pieces of

information in the sub-sections are not new and already known. Sections 3 and 4 are interrelated, so they may be merged into a single section.

Table-1 shows global primary energy use and anthropogenic heat emissions in 2018. The data is from IEA, 2020. As this is an annual snap shop, it is suggested to use a long-term climatological mean of IEA data in table-1 and subsequently revise the figure.2.

Section-6 needs to be explained in detail. Part of the heat radiated to space and atmosphere need a detailed analysis. 0.02 W/m2 was radiated into the space in 2018. It is advisable to calculate this as a mean using long-term data showing the trend of the heat radiated to the atmosphere.

Lind-338: "…..nuclear, biofuels, fossil fuel power plants emit heat which has similar global warming potential…". I guess the global warming potential mentioned here is in the sense of CO2?; if yes, then the CO2 emissions from these plants are different in nature; if not, what is the global warming potential of anthropogenic heat (excluding CO2 here)?

The author has made a good attempt to raise an important issue; however, the entire study is more theoretical in nature rather than experimental (by models and long-term observations). Most of the pieces of information given in each section are already known. Every section reads like an introduction and review of the topic.

Abstract and Conclusions are very weak; they should be both quantitative and qualitative in nature.

The present form of the study is not suitable for publication EGUsphere and may be rejected.

---

## Referee Comment (RC2)

Report for the EGUspehere manuscript titled: **'The effect of anthropogenic heat emissions on global warming.'**

The paper attempts to calculate the climate impact of energy consumption and subsequent thermal pollution in the form of sensible heat released into the atmosphere. The topic is relevant as it pertains to the role of direct heat emissions on global warming. However, the present study simply makes empirical estimates without presenting principles and fundamentals of atmospheric heat transfer. Accurate estimation of global temperatures with inclusion of heat emissions in the climate models is increasingly becoming relevant with CO2 emissions rapidly increasing. However, the paper has the scope to provide convincing scientific arguments to support the results which is missing from the manuscript in the current format.

Following the points that require further explanation:

- The methodology followed for calculating sensible heat released and subsequent temperature changes is based on data from global energy statistics using a simple three-dimensional model. A rigorous statistical analysis accounting for the radiative properties of the atmosphere should have been used as is detailed in previous studies. The three-dimensional heat distribution model used in the model rejects scientific findings of anthropogenic atmospheric heat flux based on very weak assumptions (Line 50 onwards).

- It has been established that the local heat emissions and the subsequent urban heat island effect contributes significantly to the sensible heat flux of the global atmosphere (Yang et al., 2017), while the paper claims otherwise. A proper citation of the previous and important studies that present similar global heat emission estimates is also missing in the paper (Line 85 onwards).

- The science explained is ambiguous. While the author has tried to answer public questions related to heat dynamics in the atmosphere, the scientific understanding of the process is not clearly presented. The fact that the heat released into the atmosphere is affected majorly by the atmospheric and oceanic circulation affecting its

distribution is nowhere mentioned or addressed in the paper. Block et al., 2004 reported that anthropogenic heat in regional climate models resulted in significant rise in land temperature (again contrary to what is reported in the paper). The paper also misses the latent heat released and the indirect heat emissions due to global warming.

- A proper citation of the previous and important studies (Zhang et al., 2013) that present similar global heat emission estimates is also not presented. Citations are not linked to relevant sources of information. For example, in line 43 a reference is made to the latest IPCC report but a 2013 paper is cited.

- Finally, the language used in the manuscript is not scientific with contradictory and incorrect statements provided throughout the text. Statements like, "Therefore, there is still no direct proof that carbon dioxide emissions cause global warming.", should not have been used without substantial scientific proof and reasoning as they tend to undermine global scientific effort that has established the role of CO2 in global warming as a fact. The language of the text requires to present facts more clearly and with proper citations. Also, existing knowledge on the subject presented in previous studies along with the existing gaps needs to be provided.

The paper tries to address the contribution of anthropogenic thermal emissions as a result of growing energy consumption on global rise in temperature. The quantification of anthropogenic heat released into the atmosphere is important to make more precise global climate predictions. However, the present study needs a more rigorous scientific analysis with precise presentation of the arguments. It is not recommended for publication in the present format.

References:

Block, A., Keuler, K., & Schaller, E. (2004). Impacts of anthropogenic heat on regional climate patterns. *Geophysical Research Letters*, *31*(12).

Yang, W., Luan, Y., Liu, X., Yu, X., Miao, L., & Cui, X. (2017). A new global anthropogenic heat estimation based on high-resolution nighttime light data. *Scientific data*, *4*(1), 1-11.

Zhang, G. J., Cai, M., & Hu, A. (2013). Energy consumption and the unexplained winter warming over northern Asia and North America. *Nature Climate Change*, *3*(5), 466-470.

---

## Author Comment (AC3)

Report for the EGUspehere manuscript titled: 'The effect of anthropogenic heat emissions on global warming.'

The paper attempts to calculate the climate impact of energy consumption and subsequent thermal pollution in the form of sensible heat released into the atmosphere. The topic is relevant as it pertains to the role of direct heat emissions on global warming. However, the present study simply makes empirical estimates without presenting principles and fundamentals of atmospheric heat transfer. Accurate estimation of global temperatures with inclusion of heat emissions in the climate models is increasingly becoming relevant with CO2 emissions rapidly increasing. However, the paper has the scope to provide convincing scientific arguments to support the esults which is missing from the manuscript in the current format. Following the points that require further explanation:

- The methodology followed for calculating sensible heat released and subsequent temperature changes is based on data from global energy statistics using a simple three-dimensional model. A rigorous statistical analysis accounting for the radiative properties of the atmosphere should have been used as is detailed in previous studies.

*Reply: The approach proposed in my work is innovative and, to the best of my knowledge, has not been published before. When studying a new approach, the first thing is to assess its importance under "ideal" conditions. That has been done in Lines 255-264 by showing that without heat loss from the atmosphere (mainly by radiation), anthropogenic heat release since 1850 would have resulted in the increase of atmospheric temperature by 2.3°C. The second step is to estimate the effect of anthropogenic heat release on atmospheric temperature globally under real conditions. The result is shown in Fig. 2. The next step, which is outside of the scope of this work, and will be done in the future, will be to use statistical analysis and temporal/location data. The statistical analysis done in previous studies, was based, in my opinion, on the incorrect 2-dimensional approach of anthropogenic heat release to the atmosphere. That is discussed in more details at the end of this text.*

The three-dimensional heat distribution model used in the model rejects scientific findings of anthropogenic atmospheric heat flux based on very weak assumptions (Line 50 onwards).

*Reply: I don't agree (please see below).*

- It has been established that the local heat emissions and the subsequent urban heat island effect contributes significantly to the sensible heat flux of the global atmosphere (Yang et al., 2017), while the paper claims otherwise.

*Reply: The paper of Yang, as well as many of the papers on urban heat islands, are based on the 2D approach (W/m2), which is shown below to be problematic. Regarding urban heat islands, please see my answer in Lines 303-304.*

A proper citation of the previous and important studies that present similar global heat emission estimates is also missing in the paper (Line 85 onwards).

*Reply: No citation was proposed because, to the best of my knowledge, similar 3D approach, as well as using heat directly added to the atmosphere, has not been proposed before.*

‒ The science explained is ambiguous. While the author has tried to answer public questions related to heat dynamics in the atmosphere, the scientific understanding of the process is not clearly presented. The fact that the heat released into the atmosphere is affected majorly by the atmospheric and oceanic circulation affecting its distribution is nowhere mentioned or addressed in the paper.

*Reply: As mentioned above, this is the first paper on the subject, and it uses the simplifying assumption of well mixed atmosphere (Lines 301-305).*

Block et al., 2004 reported that anthropogenic heat in regional climate models resulted in significant rise in land temperature (again contrary to what is reported in the paper). The paper also misses the latent heat released and the indirect heat emissions due to global warming.

*Reply: In my view, there ate two problems in the paper of Block: (1) the anthropogenic heat input to the atmosphere is equal to the "energy consumption", which seems to be assumed to be equal to the use of all the different primary energy sources, and (2) the heat flux is two dimensional. They are discussed below.*

‒ A proper citation of the previous and important studies (Zhang et al., 2013) that present similar global heat emission estimates is also not presented.

*Reply: Please see my reply above related to the paper of Block.*

Citations are not linked to relevant sources of information. For example, in line 43 a reference is made to the latest IPCC report but a 2013 paper is cited.

*Reply: Thanks for pointing to that mistake. During writing the paper, the latest IPCC report was released in 2013. Since the 2022 report does not address the probability of CO2 being the cause of global warming, the word "latest" will be removed from the MS text.*

‒ Finally, the language used in the manuscript is not scientific with contradictory and incorrect statements provided throughout the text. Statements like, "Therefore, there is still no direct proof that carbon dioxide emissions cause global warming.", should not have been used without substantial scientific proof and reasoning as they tend to undermine global scientific effort that has established the role of CO2 in global warming as a fact.

*Reply: My statement cited above was supported in the manuscript by the following paragraph:* "As a result, the Intergovernmental Panel for Climate Change concluded in its  report (Stocker et al., 2013) that the probability of global warming being caused by increasing atmospheric CO2 concentration is 95%. While this number seems high, it is still far from 100%. There have been many projected events of different nature with a higher expected probability that never actually occurred.*" (Lines 43-46).*

The language of the text requires to present facts more clearly and with proper citations. Also, existing knowledge on the subject presented in previous studies along with the existing gaps needs to be provided.

The paper tries to address the contribution of anthropogenic thermal emissions as a result of growing energy consumption on global rise in temperature. The quantification of anthropogenic heat released into the atmosphere is important to make more precise global climate predictions. However, the present study needs a more rigorous scientific analysis with precise presentation of the arguments. It is not recommended for publication in the present format.

*Reply: As mentioned in my replies to Reviewer 1, the manuscript will be corrected in order to improve its text.*

References:
Block, A., Keuler, K., & Schaller, E. (2004). Impacts of anthropogenic heat on regional climate patterns. Geophysical Research Letters, 31(12).
Yang, W., Luan, Y., Liu, X., Yu, X., Miao, L., & Cui, X. (2017). A new global anthropogenic heat estimation based on high-resolution nighttime light data. Scientific data, 4(1), 1-11.
Zhang, G. J., Cai, M., & Hu, A. (2013). Energy consumption and the unexplained winter warming over northern Asia and North America. Nature Climate Change, 3(5), 466-470.

*General reply: Most of the comments of the Reviewer are based on two assumptions: (1) that the 2D model of anthropogenic heat release to atmosphere, universally currently accepted, is correct; and (2) That all of the energy released by primary energy sources ("energy consumption") is entering the atmosphere. While the explanations of these points are shown in the manuscript (maybe not clearly enough), I would like to discuss them below:*

1. *The natural planetary energy fluxes (both incoming and outgoing radiation in addition to the latent and sensible heat energies) have two-dimensional nature. Therefore, they are correctly measured in 2D (W/m2). However, anthropogenic heat emissions have close to 3D nature, as they are not emitted by the planetary surface. For example, the largest by far sensible heat emitter to the atmosphere, land transport (Table 1 in the manuscript) mixes quickly the emitted heat due to the air turbulence. In addition, electrical power generation technologies plus aviation and electricity transmission, release heat high above the Earth surface (Table 1). They account for 0.96 out of a total of 5.39 Gtoe/yr (Table 1). At the same time, air mixes quickly in both horizontal and vertical direction (Lines 301-305). Therefore, a model of well mixed (3D) anthropogenic heat release was used in my paper as a first approximation (Lines 301-302). The difference between the 2D and 3D approaches is significant: while the 2D approach shows less than 1% effect of anthropogenic heat emissions on global warming, the 3D model predicts 50% effect. I am certain that 3D approach fits much better the actual process in the atmosphere. Another big question in the 2D models is: How exactly anthropogenic heat is introduced to the atmosphere? For example, Flanner (2009) states that* "heat flux was added as dry static energy to the lowest layer of the atmosphere". *When adding heat energy to an*

*object, in this case atmosphere, its temperature will rise inversely proportionally to the mass (volume) of the object (unless there is phase change, and there is none in our case). Since thickness of the atmospheric layer to which heat flux is added is not shown, the volume of the air layer is unknown and therefore, it is unknown by how much the temperature in that layer will increase. That temperature is of great importance for determining what fraction of sensible heat added to atmosphere is transformed to long wave radiation, and therefore, lost to space.*

2. *Another significant problem in the current assessment of the effect of anthropogenic heat release on atmospheric temperature is the assumption that energy released by using all the non-renewable primary energy sources ("energy consumption") goes to the atmosphere. For example, Flanner (2009) states:* "Utilizing the second law of thermodynamics, it is assumed that all non-renewable primary energy consumption is dissipated thermally in Earth's atmosphere." *However, a significant part of that energy is transformed directly to land and water (60%, according to Table 1 in my manuscript) and therefore, takes an insignificant part in heating the atmosphere (Lines 99-109).*

- *I have also a general, terminological comment. Almost always when talking about the use of different forms of primary energy in our society, the term "energy consumption" is used. A typical example if the title of the paper by Zhang et al. (2013) above. This term actually violates the first law of thermodynamics which in general states: "Energy in a closed system cannot be consumed or destroyed; it can only be converted from one form to another". In my view, the terms "energy use" or "consumption of ### energy" where "###"=electric, thermal, etc. should be used instead.*

---

## Author Comment (AC4)

**Review report of the manuscript "The effect of anthropogenic heat emissions on global warming," manuscript no. egusphere-2022-5**

This study calculates atmospheric temperature rise from 1850 to 2018 due to anthropogenic heat emissions to the atmosphere, based on a three-dimensional heat distribution approach. This study concludes that the atmosphere's temperature is half of the observed global land-sea temperature rise, and anthropogenic heat emissions significantly affect global warming.

This study raises very crucial issues but with a weak approach. The introduction section lacks a clear explanation about why this study is necessary and what are gap areas this study attempts to fill. Line 53 depicts that "there are problems in models used in the previous studies," but it does not explain the details. Line 39 says, "there is no direct proof that CO2 causes global warming" however, it does not explain this statement. Line 43 mentions the latest IPCC report but cites a 2013 paper. The introduction section should be constructively revised and clearly explain the gap areas and the detailed explanation of the need for this study.

*Reply: I agree with most of the Reviewer's suggestions above, and they will be used to correct the manuscript.*

The main assumptions of the model described in lines 62-66 are very theoretical. A details explanation is required for each assumption.

*Reply: Additional text explaining the assumptions will be added.*

Line 63 "the spatial and temporal distribution of heat and CO2 due to convection is similar in the atmosphere". Here do you mean latent heat or sensible heat?

*Reply: Here I have in mind sensible heat. That will be explained and corrected in the next version of the manuscript.*

Sensible heat is energy that moves from one system to another, changing its temperature. The transport and distribution of CO2 throughout the atmosphere are controlled by the jet stream, large weather systems, and other large-scale atmospheric circulations. A detailed explanation is required for this assumption.

*Reply: I agree, and the explanation will be added to the revised MS.*

Also, the authors consider only CO2 in their assumptions; however, black carbon is found to influence the sensible heat very differently compared to other aerosols and greenhouse gases due to its strong atmospheric absorption

*Reply: Black carbon affects the atmospheric temperature much less than $CO_2$. That is the reason not mentioning it here. New text will be added to clarify that point.*

There are confusing statements in lines 67-69, e.g., the "main difference"…." The similarities"……. Please re-write with a clear explanation.

*Reply: The text will be rewritten to avoid confusion.*

Re-arrange section 3; there are un-numbered sub-sections, e.g., precision measurements and electric power plants. It is better to merge all sub-section in section 3.

*Reply: I agree that numbering of the sections and subsections can be improved. That will be done in the corrected version of the MS.*

The pieces of information in the sub-sections are not new and already known. Sections 3 and 4 are interrelated, so they may be merged into a single section.

*Reply: In my view, the information in the sub-sections is new, as it shows the direct heat emissions to the atmosphere. To the best of my knowledge, such information does not exist in the available literature.*

Table-1 shows global primary energy use and anthropogenic heat emissions in 2018. The data is from IEA, 2020. As this is an annual snap shop, it is suggested to use a long-term climatological mean of IEA data in table-1 and subsequently revise the figure.2.

*Reply: Long-term IEA data were used in the calculations. They will be added to Table 1 in the corrected version of the MS.*

Section-6 needs to be explained in detail. Part of the heat radiated to space and atmosphere need a detailed analysis. 0.02 W/m2 was radiated into the space in 2018. It is advisable to calculate this as a mean using long-term data showing the trend of the heat radiated to the atmosphere.

*Reply: A new graph was created, showing the amount of heat radiated to space due to anthropogenic heat emissions for the period 1850-2018. It will be included in the revised version of the manuscript.*

Lind-338: "…..nuclear, biofuels, fossil fuel power plants emit heat which has similar global warming potential…". I guess the global warming potential mentioned here is in the sense of CO2?; if yes, then the CO2 emissions from these plants are different in nature; if not, what is the global warming potential of anthropogenic heat (excluding CO2 here)?

*Reply: No. The global warming potential is defined here as the increase of atmospheric temperature due to the heat emitted to the atmosphere by these industries. Table 1 shows that the atmospheric heat emitted by biofuel, oil, natural gas and coal power plants per kWh electricity generated is almost equal. In nuclear power plants that number is somehow higher, but still similar. Therefore, these industries cause a similar atmospheric temperature increase per kWh electricity generated. That will be described in the revised version of the manuscript.*

The author has made a good attempt to raise an important issue

*Reply: Thanks.*

however, the entire study is more theoretical in nature rather than experimental (by models and long-term observations).

*Reply: I would not agree. The study is based on actual numerical data for the anthropogenic energy release and on long-term observational data for atmospheric temperature chance. In my view actually model results are of theoretical nature.*

Most of the pieces of information given in each section are already known.

*Reply: The most important piece of information, the amount of direct anthropogenic heat emissions to the atmosphere, is new. It is the basis of this work.*

Every section reads like an introduction and review of the topic. Abstract and Conclusions are very weak; they should be both quantitative and qualitative in nature.

*Reply: As mentioned above, that will be rectified in the revised version of the manuscript.*

The present form of the study is not suitable for publication EGUsphere and may be rejected.

*Reply: Summarizing the above, the Referee's suggestions are almost entirely related to the revision of the text. I am ready to prepare a revised version of the manuscript, taking into account almost all of the Reviewer's comments. The important thing is that the Referee does doubt the results reported.*